# Interaction of Arsenic Exposure and Transcriptomic Profile in Basal Cell Carcinoma

**DOI:** 10.3390/cancers14225598

**Published:** 2022-11-15

**Authors:** Muhammad G. Kibriya, Farzana Jasmine, Aaron Munoz, Tariqul Islam, Alauddin Ahmed, Lin Tong, Muhammad Rakibuz-Zaman, Mohammad Shahriar, Mohammed Kamal, Christopher R. Shea, Joseph H. Graziano, Maria Argos, Habibul Ahsan

**Affiliations:** 1Institute for Population and Precision Health, Biological Sciences Division, University of Chicago Medicine, Chicago, IL 60637, USA; 2UChicago Research Bangladesh (URB), University of Chicago, Dhaka 1230, Bangladesh; 3Pulse Infoframe, London, ON N5X 4E7, Canada; 4Department of Pathology, The Laboratory, Dhaka 1205, Bangladesh; 5Section of Dermatology, Department of Medicine, University of Chicago, Chicago, IL 60637, USA; 6Department of Environmental Health Science, Mailman School of Public Health, Columbia University, New York, NY 10032, USA; 7Division of Epidemiology & Biostatistics, School of Public Health, University of Illinois Chicago, Chicago, IL 60612, USA

**Keywords:** Arsenic, Basal Cell Carcinoma, Gene expression, Gene environment interaction, DNA damage, DNA replication pathway, Hedgehog signaling, Notch signaling, Platinum drug resistance

## Abstract

**Simple Summary:**

Over 100 million individuals worldwide are exposed to arsenic through drinking water. Arsenic is a cancer-causing chemical. While ultraviolet light exposure and skin sensitivity are known risk factors for skin cancers, especially in Caucasians, arsenic exposure may be a major risk factor for other populations. This is the first study to address the genome-wide gene expression in basal cell carcinoma (BCC) of skin in the context of arsenic exposure. Our study confirms many of the gene pathways involved in BCC; in addition, we found that high arsenic exposure was associated with impaired (a) DNA replication, (b) cellular response to DNA damage repair and (c) immune response that leads to BCC. This study suggests a lower chance of platinum drug resistance in BCC patients with high arsenic exposure compared to the higher chance of platinum drug resistance in patients with low arsenic exposure. These findings may have biological and clinical significance.

**Abstract:**

Exposure to inorganic arsenic (As) is recognized as risk factor for basal cell carcinoma (BCC). We have followed-up 7000 adults for 6 years who were exposed to As and had manifest As skin toxicity. Of them, 1.7% developed BCC (males = 2.2%, females = 1.3%). In this study, we compared transcriptome-wide RNA sequencing data from the very first 26 BCC cases and healthy skin tissue from independent 16 individuals. Genes in “ cell carcinoma pathway”, “Hedgehog signaling pathway”, and “Notch signaling pathway” were overexpressed in BCC, confirming the findings from earlier studies in BCC in other populations known to be exposed to As. However, we found that the overexpression of these known pathways was less pronounced in patients with high As exposure (urinary As creatinine ratio (UACR) > 192 µg/gm creatinine) than patients with low UACR. We also found that high UACR was associated with impaired DNA replication pathway, cellular response to different DNA damage repair mechanisms, and immune response. Transcriptomic data were not strongly suggestive of great potential for immune checkpoint inhibitors; however, it suggested lower chance of platinum drug resistance in BCC patients with high UACR compared high platinum drug resistance potential in patients with lower UACR.

## 1. Introduction

Skin cancers are divided into two groups—(a) Melanoma, when cancer arises in melanocytes and (b) Non-melanoma skin cancer (NMSC), when cancer arises in squamous, basal, or Merkel cells. NMSC is the most common type of skin cancer and has shown a rise in incidence worldwide. The two main histological types of NMSC are basal cell carcinoma (BCC) and squamous cell carcinoma (SCC). Changes in the basal cells could lead to abnormal and uncontrolled growth leading to BCC. BCC constitutes approximately 75% of NMSC [1]. BCC is observed mostly in the neck (80% of the cases), followed by the trunk (15% of the cases), and represents 80% of all cases of NMSC with an incidence of approximately 2 million new diagnoses each year in the United States [2,3]. Skin cancer is less common in persons with darker skin pigmentation compared with light-skinned Caucasians but is often associated with greater morbidity and mortality [4]. Although, compared with Caucasians, BCC is lower in the Asian population, but the incidence is increasing there as well [5].

Etiology of BCC is a multifactorial combination of genotype, phenotype, and environmental factors. Multiple risk factors have been described for BCC, but chronic exposure to ultraviolet (UV) radiation, particularly UVB, is clearly the most important [6], although it is more strongly associated with SCC than BCC [7]. NMSC, especially BCC, is the most common cancer affecting white-skinned individuals and the incidence is increasing worldwide [8,9]. Asians develop BCCs later in life and develop fewer BCCs over their lifetime than Caucasians, despite similar estimated lifetime daily sun exposure. Authors concluded that this is probably due to skin pigmentation in Asians being more protective of UV light than skin pigmentation in Caucasians [10].

Exposure to inorganic Arsenic (As) has long been recognized as a risk factor for BCC through ingestion of As-contaminated water [2]. BCC are linked to As exposure and develops both in sun-exposed skin and covered parts [11]. As is a naturally occurring element that is found in combination with either inorganic or organic substances to form many different compounds. Inorganic As compounds are found in soils, sediments, and groundwater. Organic As compounds are found mainly in fish and shellfish. In the past, inorganic forms of As were used in pesticides and paint pigment. People are most likely to be exposed to inorganic As through drinking water and, to a lesser extent, through various foods. Inorganic As and As compounds are considered to be cancer-causing chemicals.

Genetic component also plays an important role in BCC development. Most of the studies addressing the molecular changes in BCC are carried out through either the candidate gene approach or in vitro studies [2,7,12,13,14]. There are few studies on DNA copy number changes in BCC [15,16].

To our knowledge, no study was done at the genome-wide level to address the effect of As exposure on gene expression in BCC. Our group has been following up one of the largest cohorts exposed to different levels of As through drinking water [17]. In a prospectively followed-up large cohort exposed to As, we found that male gender, higher age, higher level of As exposure, and genomic deletions in several genes and lincRNA genes may predispose an individual to a higher risk of development of As-induced skin lesions [18]. In Bangladesh Vitamin E and Selenium Trial (BEST), our group conducted a double-blind placebo-controlled study, to evaluate the effect of vitamin E and selenium in the prevention of NMSC in a population exposed to As who had clinical manifestation of As toxicity in the form of As-related non-malignant skin lesion [19]. Utilizing clinical samples from the BEST for RNA sequencing, in this paper, we studied the transcriptomic profiling of BCC to explore the association of level of As exposure and BCC at the gene expression level.

## 2. Materials and Methods

For this study, we used the clinical data and samples from the BEST study. The study population included 7000 men and women (m = 2840, f = 4160) who were known to be exposed to As through drinking well water contaminated with naturally occurring As. All of them had clinically visible non-malignant skin lesion (melanosis or leukomelanosis or keratosis)—a known manifestation of As toxicity. As a part of the clinical trial, they were actively followed up for six years for the development of NMSC. Urinary As creatinine ratio (UACR) at baseline and at 2-year, 4-year, and 6-year follow-up visits showed persistence of As exposure. Skin biopsy was performed in all patients who had reasonable clinical suspicion of Bowens Disease (BD) or NMSC, including SCC and BCC, and they consented for biopsy. Thus, 14.7% of the males and 7.5% of the females (*p* < 0.0001, Chi-square test) had skin biopsy done in the course of the 6-year follow-up. 

Histopathology examination was performed on Hematoxylin and Eosin (H&E) stained slides from routinely processed paraffin impregnated tissue blocks. Two pathologists examined the slides independently. For the pathological diagnosis, a structured reporting form was used (see Form S1). Out of total 727 unique participants (m = 417, f = 310) who had skin biopsy, 37.7% of the biopsies showed BD, 2.9% had invasive SCC and 16.1% showed BCC and the rest 43.3% did not show any histological evidence of NMSC. These non-NMSC lesions consisted mostly of Arsenical keratosis and other skin conditions. Histology slide photomicrograph of each type of lesion are shown in Figure 1. 

Figure 1A shows Basal Cell Carcinoma: Collections of atypical cells budding from the epidermal basal layer, with peripheral palisading and myxoid stroma. Figure 1B shows invasive Squamous cell carcinoma: Collections of atypical keratinocytes infiltrating into the dermis. Figure 1C shows Bowen Disease (Squamous cell carcinoma in situ): Epidermis shows hyperkeratosis, acanthosis (thickened spinous layer), and numerous atypical keratinocytes, present throughout the full thickness of the epidermis. Figure 1D shows Arsenical keratosis: Epidermis shows hyperkeratosis, acanthosis (thickened spinous layer), and scattered atypical keratinocytes, confined to the lower half of the epidermis.

If we assume that NMSC was not missed from among the population who did not require a skin biopsy (n = 6273), then from among these As-exposed study population, 2.2% of the male and 1.3% of the female participants developed BCC, while 0.4% of the male and 0.2% of the female participants developed SCC over the six-year follow-up. The detailed result from this trial will be reported in a separate paper (manuscript in preparation).

We used UACR at baseline and 2-year, 4-year and 6-year follow-up as a measure of As exposure. The urinary total As concentration was measured by graphite furnace atomic absorption spectrometry [20]. Urinary creatinine was measured by a colorimetric method based on the Jaffe reaction described by Heinegard and Tiderstrom [21]. The urinary As was measured from a spot urine collection. To take into account the hydration status, we used the UACR. Based on our previous genomic study in general population from the same geographical area [18], we dichotomized the UACR values of ≤192 µg/gm creatinine as lower (coded as 0) and >192 µg/gm creatinine as higher (coded as 1). It may be noted that this cutoff of UACR roughly corresponds to drinking water As level of ~50 µg/L, which is five times higher than the highest acceptable contamination level according to the WHO guideline (10 µg/L). We used the same cut-off value of UACR for baseline, 2-year, 4-year, and 6-year follow-up for categorization. At baseline, 69.4% participants had high As exposure (UACR ≥ 192 µg/gm creatinine). For the total study population, this proportion was 64.0% at 2-year, 59.6% at 4-year follow-up, and 56.6% at 6-year follow-up without any difference between males and females. This shows that despite attempt of public awareness and some mitigation effort, unfortunately, the vast majority of the participants had high As exposure throughout the study period. Among these participants having visible As toxicity with comparatively lower As exposure (UACR baseline ≤ 192 µg/gm creatinine) at baseline, 1.6% developed BCC, and among the participants with comparatively higher As exposure (UACR baseline > 192 µg/gm creatinine) at baseline, 1.7% developed BCC during the 6-year follow-up.

In this molecular study, we have included the very first 26 biopsy confirmed cases of BCC for which we have tumor tissue properly preserved in RNA later and compared with similarly preserved healthy skin tissue from biopsy material, resected as the surrounding tissue from first 16 independent patients with Arsenical keratosis (see Appendix A). In general or combined group, the BCC were more frequently seen in the head & neck region than the “healthy skin” used for comparison (*p* = 0.051), but it was not statistically different either in male (*p* = 0.135, chi-square test) or in female subjects (*p* = 0.317, chi-square test) separately due to small sample size. The distribution of location of the samples among the low UACR group (≤192 µg/g creatinine) and high UACR group (>192 µg/g creatinine) also were not different. In this paper, we have focused only on the transcriptomic profile of BCC and therefore did not present the detail data on the other lesions (e.g., BD, SCC and arsenical keratosis) throughout the follow-up period. The detailed result including the other lesions from the BEST trial will be reported in a separate paper (manuscript in preparation).

RNA was extracted from these RNA later preserved tissues using Quick-DNA/RNA Microprep Plus kit (Zymo Research, Irvine, CA, USA) following manufacturer’s protocol. RNA Quantification and 260/280 ratio was checked by NanoDrop 1000. For RNA sequencing on Illumina platform, we used Lexogen’s Quantiseq 3’ mRNA–Seq kit (Vienna, Austria) for library preparation. This protocol generates only one fragment per transcript, very close to 3’ of the transcript. The input RNA was 40 ng per well. An oligo-dT primer containing illumina-compatible sequence at the 5’ end was used for RNA hybridization and reverse transcription. In the next step, the RNA template was degraded and removed. After that step, second strand synthesis was done using random primer containing Illumina-compatible linker sequence at the 5’- end. At this step, the double stranded DNA was formed. This ds-DNA was then purified using magnetic beads. This library is then amplified to add the complete adapter sequences required for cluster generation. Then, magnetic purification was done to clean up the library by removing the PCR reagents.

The final library was measured by fluorometer, and after pooling, qPCR was done to quantify the input library for sequencing on the Illumina HiSeq platform (San Diego, CA, USA).

### Statistical Analysis

The RNA sequence data was processed using Partek Flow (version 10.0) (https://www.partek.com/partek-flow/, accessed on 11 November 2022). STAR aligner was used for alignment, and the final gene count data was expressed as count per million reads (CPM) and were log_2_ transformed for the ANOVA using Partek Genomics Suite (version 7.0) (https://www.partek.com/partek-genomics-suite/, accessed on 11 November 2022). Using ANOVA models including an interaction term (Tissue × Factor), we examined if the gene expression in tumor tissue (0 = healthy, 1 = BCC) was significantly different in the presence or absence of any particular factor (e.g., As exposure). Factors were dichotomized as 0 or 1.

In GO Enrichment analysis, we tested if the genes found to be differentially expressed fell into a Gene set (defined by KEGG pathway or other gene sets of interest) more often than expected by chance. We used a chi-square test to compare “number of significant genes from a given gene set/total number of significant genes” vs. “number of genes in that gene set that have been sequenced/total number of genes on the sequencing data”. Negative log of the *p*-value for this test was used as the enrichment score. Therefore, a Gene set with a high enrichment score represents a lead functional group. The enrichment scores were analyzed in a hierarchical visualization and in tabular form.

Gene set ANOVA is a mixed-model ANOVA to test the expression of a set of genes (sharing the same category or functional group) instead of an individual gene in different groups [22]. The analysis is performed at the gene level, but the result is expressed at the level of the Gene set category by averaging the member genes’ results. The equation for the model was:Model: Y = μ + T + P + G + S (T × P) + ε
where Y represents the expression status of a Gene set category, μ is the common effect or average expression of the Gene set category, T is the tissue-to-tissue (tumor/normal) effect, P is the patient-to-patient effect, G is the gene-to-gene effect (differential expression of genes within the gene set category independent of tissue types), S (T × P) is the sample-to-sample effect (this is a random effect and is nested in tissue and patient), and ε represents the random error.

## 3. Results

The diagnosis of BCC was confirmed by pathological biopsy as shown in Figure 1. There was consensus between two pathologists for all the 26 BCC cases included in this study.

### 3.1. Differential Gene Expression in BCC at Gene Level:

Differential expression analysis from the sequencing data suggested that 136 genes were differentially expressed by at least 2-fold at FDR 0.05 level (see Appendix A for detail). The enrichment analysis of this list of differentially expressed genes is shown in Appendix A. It was seen that the genes involved in “Basal cell carcinoma” pathway, and “Hedgehog signaling” pathways were over-represented (enriched) in this list. These pathways are also found in other studies addressing BCC in other population [21,22,23].

### 3.2. Differential Gene Expression in BCC at Pathway Level

In the next step, instead of an individual gene, we examined if a set of genes (sharing the same category or functional group, e.g., in KEGG pathway) was differentially expressed in BCC tissue compared to healthy skin tissue. We used the Gene set ANOVA. The detailed result is shown in Appendix A. This analysis also showed that among the top differentially expressed pathways in BCC, are “Hedgehog signaling pathway” (see Figure 2) and “Basal cell carcinoma pathway” (see Figure 3). This is consistent with the above finding of the enrichment analysis of list of individual genes differentially expressed in BCC. On average, genes in these pathways were ~2.2–2.3-fold over-expressed in BCC tissue compared to healthy skin tissue independent of gender, age group (≤38 years vs. >38 years), BMI category (≤18 vs. >18) and As exposure at baseline (UACR ≤ 192 µg/gm creatinine vs. UACR > 192 µg/gm creatinine). We included these factors in the Gene set ANOVA models because in our previous study addressing As-related skin lesion as well as in the parent study BEST, these were significantly associated with clinically manifest skin lesion or histologically proven skin cancer.

### 3.3. Level of As Exposure and Differential Expression of Genes

In the next step, we tried to identify if the magnitude of differential expression of some gene set(s) was different among the patients with lower As exposure compared to those with higher As exposure. We included an interaction term “BCC × UACR_baseline_” in the Gene set ANOVA model(s); the *p*-value of that interaction term identified pathways/gene sets that were differentially expressed in BCC depending on level of UACR. It was interesting to note that the magnitude of differential expression of genes in some of the well-known pathways involved in BCC were different among patients with low and high As exposure at baseline. “Notch signaling pathway” (see Figure 4), “Basal cell carcinoma pathway” (see Figure 5), and “Hedgehog pathway” (see Figure 6) are a few of them. The detailed list is shown in Appendix A. Similarly, we also looked at UACR at 2-year, 4-year, and 6-year follow-up levels to see if this effect was replicated (data not shown), and we found similar results suggesting that at higher level of As exposure (at baseline and follow-up), the magnitude of differential expression (BCC vs. healthy skin tissue) of these set of genes were smaller.

Another important difference was in the genes involved in the “DNA replication pathway” (see Figure 7). Our result suggested that the DNA replication pathway was impaired in high arsenic exposure.

### 3.4. Level of As Exposure and Differential Expression of Selected Cancer Related Gene-Sets

In the light of well-known biological processes involved in carcinogenesis, we examined if the level of As exposure was associated with differential expression of some selected group of genes, involved in different molecular mechanisms, known to be altered in cancer—such as– like DNA repair, apoptosis, caspase initiator, caspase executioner, tumor suppressor, anti-tumor suppressor, etc. (see Appendix A for a complete list of genes we tested). Using the interaction term “BCC × UACR_baseline_” in Gene set ANOVA, we detected the functional group(s) of genes where the magnitude of differential expression was different in patients with lower and higher UACR at baseline. The difference of differential expression (Fold Change) in the low and high UACR group patients is shown in Table 1.

We found that the most significant group was “DNA repair” (ANOVA interaction *p* = 1.84 × 10^−6^). In the low UACR group, DNA repair genes were 2.79-fold (95% CI 1.98–3.93) over-expressed in BCC tissue compared to healthy skin tissue, whereas in the high UACR group, these DNA repair genes were not even significantly overexpressed compared to healthy skin tissue [1.01-fold change (95% CI −1.26–1.27, *p* = 0.96); see Figure 8]. In fact, compared to BCC in low UACR, the BCC tissue in high UACR showed significant down regulation of DNA repair genes [2.53-fold change (95% CI from −1.93 to −3.31), *p* = 3.6 × 10^−11^]. These gene expression data suggest that high As exposure is associated with impaired DNA repair in BCC. The differences in other cancer-related gene sets are shown in Table 1.

### 3.5. Level of As Exposure and Differential Expression of Selected DNA Damage Related Gene-Sets

In the next step, we further explored if differential expression of any particular group or type of DNA-damage-related genes were different among the BCC tissues depending on low or high As exposure. Most of the biological processes involved in DNA damage repair were less activated in patients with high As exposure (see Table 2 for detail) compared to those with low As exposure. Genes involved in translesion synthesis (TLS) were over-expressed in BCC by 3.67-fold (95% CI 2.34–5.74) among low UACR group (see Figure 9), whereas among the high UACR group, this group of genes were not even differentially expressed [1.14-fold change (95% CI −1.19–1.55))] in BCC compared to healthy skin tissue. In fact, compared to BCC among low As exposure, TLS genes were down regulated [2.45-fold change (95% CI from −1.72 to −3.49), *p* = 8.23 × 10^−7^] in BCC among high As exposure. Translesion synthesis is a DNA damage tolerance mechanism that allows for bypass of DNA damage during replication; this prevents replication fork breakdown and genomic instability. Translesion synthesis polymerases are required to tolerate DNA lesions that otherwise cause replication arrest and cell death.

Genes involved in “nucleotide excision repair” (see Figure 10), “base excision repair”, “direct reversal repair”, “micro homology mediated end joining”, “non-homologous end joining”, “mismatch repair” as well as “checkpoint signaling” all showed significant up-regulation in BCC only in low As exposure but no change in the presence of high As exposure. Only the “homologous recombination” genes were slightly up regulated in BCC [1.41-fold change (95% CI 1.04–1.92)] in high As exposure, but again the fold change was significantly lower (interaction *p* = 0.02) than the up regulation seen in BCC among low As exposure [2.57-fold change (95% CI 1.64–4.03)].

### 3.6. Level of As Exposure and Differential Expression of Replication Stress Gene-Sets

In the light of the above results of association of As exposure and DNA damage repair machinery, we explored if the genes involved in different biological processes in replication stress are different depending on level of As exposure. The total list of genes at replication stress site and their functional group that we tested is shown in Appendix A. Analyses of our data (see Table 3) suggested that differential expression of genes (BCC vs. healthy) involved in all the biological processes related to the replication stress response was attenuated in high As exposure. In fact, most of these functional groups of genes were down regulated in high- As- exposure condition. Figure 11 shows the example of impaired replication stress response of “DNA replication repair” genes in higher UACR group.

### 3.7. Gene Expression of Tumor Tissue in Low and High As Exposure

We also compared the gene expression in BCC tissue of patients having high baseline As exposure (UACR baseline > 192 µg/gm creatinine) to BCC tissue from patients with comparatively lower UACR (UACR baseline < 192 µg/gm creatinine). We found that a total of 38 genes were at least 2-fold differentially expressed at FDR 0.05 level in BCC from the high-exposure group (for details, see Appendix A). All of them were down-regulated in high UACR group. The enrichment analysis of this list of down regulated genes is shown in in Figure 12. It was seen that the list was enriched in genes involved in “Immune response” and “leukocyte activation”, indicating that higher As exposure decreases the immune response.

### 3.8. Level of As Exposure and Potential Therapeutic Implication

All the cases of BCC in our study were treated with surgical excision and none was treated or needed any anti-cancer chemotherapy. However, we explored the RNA sequencing data for potential use of immune checkpoint inhibitors. PD-L1 (also known as CD274) was only marginally overexpressed (*p* = 0.02) in BCC compared to healthy skin tissue in the low-As-exposure setting, suggesting potential beneficial effect, but PD-L1 was not overexpressed in tumor tissue in the high-As-exposure group (*p* = 0.83) possibly suggesting lack of response to PDL1 inhibitors in such cases. We did not find any difference of differential expression of the other immune target genes (e.g., LAG3, CTLA4, *or HAVCR2*) in BCC compared to healthy skin tissue in the presence or absence of high As exposure that could suggest the potential use of other immune checkpoint inhibitors targeting those genes. However, it was interesting to note that genes related to platinum drug resistance were significantly overexpressed in BCC only in the relatively lower As-exposure group but not in the presence of high As exposure (see Figure 13). This finding may suggest that although the cis-platin may not be effective or develop resistance in BCC with low As exposure, but it may be effective or may have lower chance of developing drug resistance in BCC when it develops in the presence of a higher As-exposure condition. The detailed expression data of all the genes related to platinum drug resistance is presented in Appendix A. Among the genes that showed significant interaction with As exposure, some of them are worth mentioning. Compared to healthy skin tissue, REV3L was overexpressed by 3.83-fold (95% CI 1.97–7.45; *p* = 2.15 × 10^−4^) in the low As exposure group as compared to just 1.17-fold (95% CI −1.35–1.86, *p* = 0.488) in the high As exposure group. Similarly, compared to healthy skin tissue, *POLH* expression increased by 4.42-fold (95% CI 1.18–16.64; *p* = 2.88 × 10^−2^) in the low As exposure group and decreased 1.37-fold (95% CI from −3.44 to 1.83); *p* = 0.491) in the high-As-exposure group.

## 4. Discussion

Over 100 million individuals worldwide are exposed to As through drinking water, including 28–57 million in Bangladesh and 2.1 million in the United States [23]. Chronic As exposure through drinking water is associated with an increase in mortality [24]. Most As-related cancers have a long latency period, but As-induced skin lesions appear relatively early [25,26]. While UV light exposure and skin sensitivity are known risk factors for NMSC, especially in Caucasians, As exposure through contaminated drinking water may be a major risk factor for other populations [27,28,29].

To our knowledge, this is the first study to address the genome-wide gene expression in BCC in the context of As exposure. We could not compare BCC developing in an As-free environment and BCC developing among As-exposed individuals because the parent study from where we used the tissue samples we used, was conducted in a population already exposed to As via drinking As-contaminated well water and already had clinically manifest As related skin lesion. However, we had a large variation of As exposure among the studied population that allowed us to compare the effect of low and high As exposure on the transcriptomic profile of BCC. We also acknowledge the fact that it would have been ideal to have the normal skin tissue from the same individual. However, unfortunately, the corresponding adjacent healthy skin tissue was not available for RNA extraction. We also acknowledge the weakness that we did not measure the As content in the tissue in this study. In fact, from the surgical excision biopsy, the major portion of the tissue was used for pathology (FFPE block) and only a small portion was preserved in RNA later for DNA RNA-based molecular testing. The whole fraction was used up for DNA and RNA extraction and we did not have any tissue left for mass spectroscopic estimation of As content in the tissue. We totally understand the significance of measuring the As content in the tissue to answer the research question from etiological point of view and hopefully will collect tissue sample accordingly for future study. In previous study from same region, we observed a significant dose response relation of As exposure and the As skin toxicity. Compared with drinking water containing <8.1 μg/liter of As, drinking water containing 8.1–40.0, 40.1–91.0, 91.1–175.0, and 175.1–864.0 μg/liter of As was associated with adjusted prevalence odds ratios of pre malignant skin lesions of 1.91 (95% confidence interval (CI): 1.26, 2.89), 3.03 (95% CI: 2.05, 4.50), 3.71 (95% CI: 2.53, 5.44), and 5.39 (95% CI: 3.69, 7.86), respectively [17]. That result led us to think that As exposure may be associated with high incidence of NMSC in this population. Although the current study was not designed to address the research question of effect of As exposure on the development of BCC, it was fortunate that in the actual follow-up of these subjects over six years, we did not see more that 1.7% of them developed BCC. We do not have data from people not at all exposed to As, and therefore cannot comment on causality or compare the incidence. But 1.6% of 700 individuals developing BCC over 6 years translates to approximately 266 new cases per 100,000 person years in this population. 

Many of our genome-wide findings are consistent with previously conducted candidate-gene-approach-based studies that explored gene expression status in BCC. In our study, genes in “basal cell carcinoma pathway”, “Hedgehog signaling pathway”, and “Notch signaling pathway” were overexpressed in BCC, confirming the findings from earlier studies in BCC in other population not known to be exposed to As. There are studies that showed the significance of sonic hedgehog pathway in BCC [30,31]. Similar results were also shown in animal model [32,33]. Previous gene expression study on BCC tissue showed *PTCH1* mRNA levels were upregulated [34]. In another study, the expression profile of HH-signaling-related molecules, *GLI1*, *GLI2*, *PTCH1*, *PTCH2*, *SHH*, and *SMO* in BCC and various other cutaneous tumors were examined by real-time PCR analysis and demonstrated that BCC showed remarkably enhanced mRNA expression of all HH molecules, except SMO, compared to other skin tumors [35]. Chromogranine A *(CHGA*), Chondroitin sulfate proteoglycan 4 (*CSPG4*) have been overexpressed in NMSC [36,37].

In a study, IHC was performed in a tumor and the adjacent skin tissue from the patient with BCC [31]. Similar to our findings, comparison between BCC and adjacent healthy tissue revealed that *GLI1*, *SMO*, and insulin-like growth factor 2 mRNA-binding protein 1 (*IGF2BP1*) were overexpressed [31]. Another study showed that *HMGA1* and *MMP-11* mRNA and protein expression patterns were higher in NMSC compared with normal skin [38]. Pompei V et al. performed immunohistochemical test on 79 FFPE tissue (BCC40 and SCC 39) and found *NNMT* (Nicotinamide N-methyltransferase) to be overexpressed in BCCs compared with control tissues [39]. Neinaa YME et al. compared skin tumor with healthy skin tissue by immunohistochemical staining, and significant *SOX18* overexpression was observed in cutaneous tumors in SCC and BCC. [40]. Early events of BCC tumorigenesis are triggered by inappropriate activation of SHH signaling. Early events of BCC tumorigenesis are triggered by inappropriate activation of SHH signaling, via the loss of Patched1 (*Ptch1*) or by activating mutations of Smoothened (*Smo*) [41]. *TBX1*, a transcription factor, is connected to several major signaling systems, such as FGF, WNT, and SHH, and it has been linked to cell proliferation and to the regulation of cell shape and cell dynamics. One study showed that *TBX1* was expressed in all of the 51 BCC samples that they tested, while in healthy human skin, it was only expressed only in the hair follicle [41].

Boonchai W et al. found that *TP53* gene expression in As-related BCC was less frequent and with less intensity than sporadic BCC. The BCCs from sun-exposed sites, whether As-related or sporadic, more frequently showed overexpression of *p53* than those from less-exposed areas (*p* = 0.004; 2-tailed test). The more aggressive subtypes of BCC showed a higher level of expression of p53 than the less aggressive forms [7].

Bonifas JM et al. [42] tested the gene expression level of 27 fresh frozen BCC tissue by qPCR. Compared to normal cultured cell, BCC cells have increased levels of mRNA for *PTC1*, *GLI1*, *HIP*, *WNT2B*, and *WNT5a*; decreased levels of mRNA for *c-MYC*, *c-FOS*, and *WNT4*; and unchanged levels of mRNA for *PTC2*, *GLI2*, *WNT7B*, and *BMP2* [42]. The Xeroderma pigmentosum complementation group C protein (*XPC*), a general sensor of damaged DNA, was significantly higher in BCC compared to adjacent normal epidermis [43].

Using tissue microarrays Young LC et al. examined both nuclear and cytoplasmic levels of MMR proteins *MSH2*, *MSH6*, *MSH3*, *MLH1*, and *PMS2* in FFPE tissue in more than 200 cases of cutaneous SCC and BCC by immunohistochemistry [44]. They found that subsets of these 10 MMR protein measures were increased in NMSC compared with normal epidermal samples, particularly in SCC. With the exception of nuclear *MSH2*, the BCC had lower levels of identified MMR protein measures than SCC. Similar to our finding, Sahu A et al. also found higher *PARP1* expression in 95 BCC tissues using immunohistochemistry [45].

Biray Avci C et al. examined 34 pair healthy and BCC tissue and *EGFR* expression was investigated by RT PCR, and *EGFR* was found to be expressed at a significantly higher rate in BCC tissue compared to healthy tissues (*p* < 0.05) [46]. In patients with recurrence lesions, *EGFR* expression was 6.66 times higher compared to patients with non-recurrent lesions. In addition, there was a statistically significant difference of *EGFR* expression for infiltrative subtypes (*p* < 0.05) [46]. Depending on histopathological subtypes, the immunoexpression of the different gene markers may vary. Another study analyzed the immunoexpression of *EGFRs* in 53 cases of nodular, adenoid, and morpheaform BCCs by Immunohistochemistry [47]. They found significant differences in the expression of *EGFR*, *HER2*, and *HER3* reported to histological BCC types. The nodular type presented the weakest expression of *EGFR*s, while the morpheaform type had a high expression of all receptors, and the adenoid type had an increased expression only in the case of *EGFR* and *HER2* [47].

In our present study, after we looked at the differential expression between BCC and healthy skin tissue as a whole, in the next step, we wanted to see if the magnitude of differential expression was influenced by the level of As exposure in these clinical samples. There are studies on cultured cells where the effect of As was investigated on different genes. As exposure was found to be associated with *ERCC2* overexpression mostly in cases with skin lesion due to hypomethylation of its promoter region [48]. Wang A et al. [49] investigated the effect of dimethylarsinic acid (DMA) on urinary bladder transitional cultured cell lines by RT PCR and showed the mRNA levels of tested DNA repair genes, ataxia telangectasia mutant (*ATM*), X-ray repair cross-complementing group 1 (*XRCC1*), excision repair cross-complementing group 3/xeroderma pigmentosum B (*ERCC3/XPB*), and DNA polymerase b (Polb) were not altered [49]. Ding X et al. showed that arsenite interferes with *PARP-1* and *XPA* binding to chromatin and that zinc supplementation fully restores DNA binding activity of both proteins [50].

Our study shows that most of the biological processes involved in DNA damage repair were less activated in patients with high As exposure. The list includes genes involved in “translesion synthesis”, “nucleotide excision repair”, “base excision repair”, “direct reversal repair”, “micro homology mediated end joining”, “non-homologous end joining”, “mismatch repair” as well as “checkpoint signaling”. Some studies showed that Arsenite and its metabolite monomethylarsonous acid (MMA(III)) strongly decreased expression and protein level of Xeroderma pigmentosum complementation group C (*XPC*), which is believed to be the principle initiator of global genome nuclear excision repair [51,52]. There are studies which showed that As decreases cell growth in vitro that is associated with down regulation of *CDK*. Kim JY et al. examined possible roles of *p38MAPK* in the sodium arsenite-induced cell growth inhibition in NIH3T3 cells [53]. Sodium arsenite caused transient cell growth delay with marked activation of p38MAPK. In addition, arsenite induced *CDK* inhibitor *p21* (*CIP1*/*WAF1*) and enhanced its binding to the *CDK2*, which resulted in inhibition of *CDK2* activity [53]. In another study, cells treated with acute exposure of As exhibited a decrease in viability and changes in morphology. In contrast, during 24 weeks of As exposure, the cells had increased EGFR expression and activity and increased mRNA and protein levels of *TGFα* [54]. We found overexpressed *EGFR* in BCC both in high and low As exposure. Andrew AS et al. analyzed expression of DNA repair genes in lymphocytes by qPCR after As exposure in a subset of 16 individuals enrolled in a population-based case–control study investigating As exposure and cancer risk in New Hampshire [55]. Toenail As levels were inversely correlated with expression of critical members of the nucleotide excision repair complex, *ERCC1* (r^2^ = 0.82, *p* < 0.0001), *XPF* (r^2^ = 0.56, *p* < 0.002), and *XPB* (r^2^ = 0.75, *p* < 0.0001) [55]. Our finding supports that previous finding.

Suetomi K et al. compared the gene expression profiles between cells, treated and untreated and treated with 1 μM sodium arsenite (NaAsO_2_). Hundreds of genes appeared upregulated and downregulated more than two-fold, 2 h after the treatment, including *HMOX1, INHBA,* and *ANKRD11* [56]. This [56]. It supports our findings. Stronger expression of *CDK-2* has been shown in BCC and *SCC* [57,58].

In our study, the interaction of As exposure and the genes related to platinum drug resistance may allow us to consider the use of cis-platin in BCC among the patients with high As exposure. Xie K et al. found that the emergence of drug resistance to cisplatin has been linked to REV3L and *REV1* activity in murine models of B-cell lymphoma and lung adenocarcinoma; decreasing expression of *REV1* or *REV3* via short hairpin RNA significantly sensitized these tumors to platinum-based treatment and stifled the emergence of drug resistance [59]. During their retrospective analysis of squamous cell carcinomas of head and neck (HNSCC), Zhou and colleagues reported that low *POLH* expression was significantly associated with high complete response rate (*p* = 0.03) in patients treated with platinum- and gemcitabine-based chemotherapy [60]. Sokol AM et al. observed a positive relationship between *POLH* expression and DNA replication in human cells treated with DNA-damaging platinum-based drugs: carboplatin and cisplatin [61].

## 5. Conclusions

Our transcriptome-wide RNA sequencing data shows the interactions of As exposure and gene expression profiling in BCC in an Asian population exposed to inorganic As through drinking As-contaminated well water. The results confirm many of the previously known pathways involved in BCC and additionally suggests possible association of As exposure and impairment in DNA damage repair and DNA replication machinery in the pathogenesis of BCC. The data also suggested a molecular basis of lower chance of platinum drug resistance in BCC patients with high As exposure compared to the higher chance of platinum drug resistance in patients with low As exposure. These findings may have potential biological and clinical significance.

## Figures and Tables

**Figure 1 cancers-14-05598-f001:**
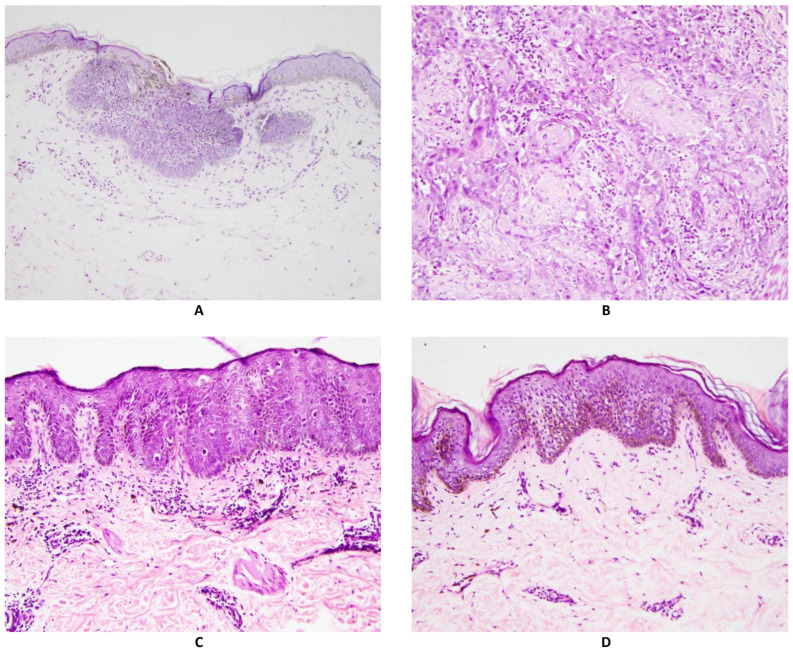
Photomicrographs of different skin biopsy findings. All are Hematoxylin and eosin stain, magnification 200×.

**Figure 2 cancers-14-05598-f002:**
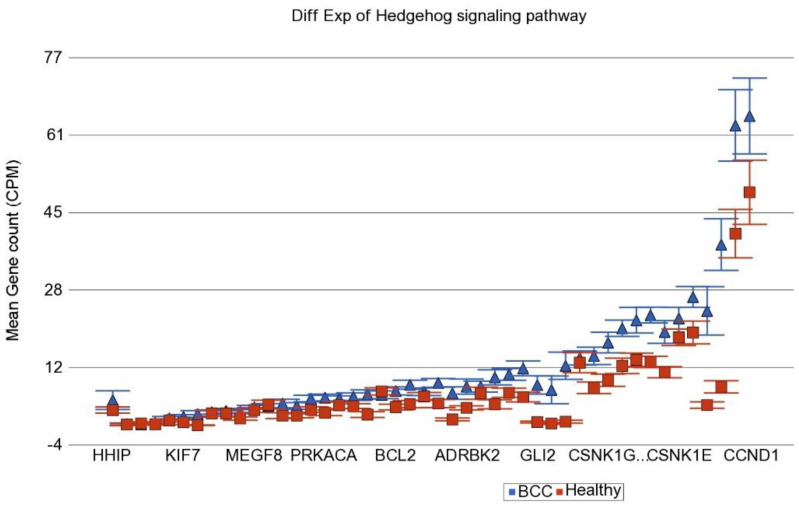
Differential gene expression of Hedgehog signaling pathway genes in BCC tissue (in blue) compared to healthy skin tissue (in red). Genes are arranged on the x-axis by expression level, and the mean gene count (counts per million) value is shown on the y-axis. Gene symbols for all of the genes could not be shown on the x-axis.

**Figure 3 cancers-14-05598-f003:**
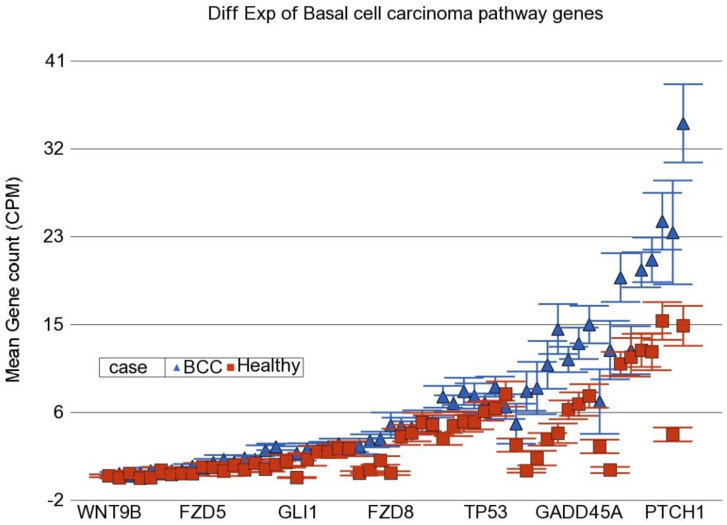
Differential gene expression of basal cell carcinoma pathway genes in BCC tissue (in blue) compared to healthy skin tissue (in red). Genes are arranged on the x-axis by expression level, and the mean gene count (counts per million) value is shown on the y-axis. Gene symbols for all of the genes could not be shown on the x-axis.

**Figure 4 cancers-14-05598-f004:**
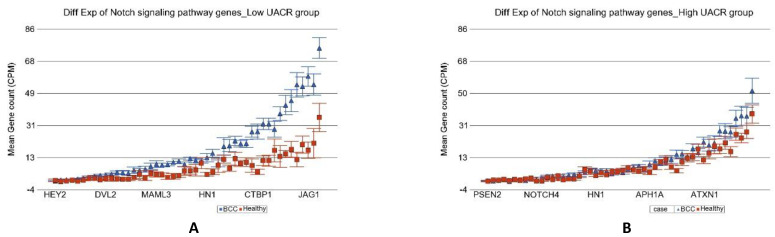
Differential gene expression of Notch signaling pathway genes in BCC tissue (in blue) compared to healthy skin tissue (in red). Genes are arranged on the x-axis by expression level, and the mean gene count (counts per million) value is shown on the y-axis. Gene symbols for all of the genes could not be shown on the x-axis. Data from patients from the low As exposure group are shown on the left (**A**), and data from patients from the high As exposure group are shown on the right (**B**). The average magnitude of overexpression was significantly higher (*p* = 4.60 × 10^−16^) if the patient had low As exposure [3.92-fold change (95% CI 3.07–5.01)] compared to those with high As exposure [1.14-fold change (95% CI −1.04–1.35)].

**Figure 5 cancers-14-05598-f005:**
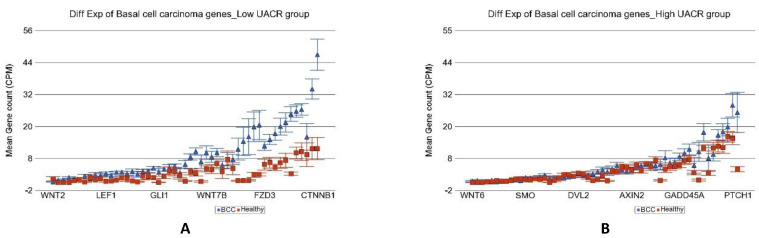
Differential gene expression of basal cell carcinoma pathway genes in BCC tissue (in blue) compared to healthy skin tissue (in red). Genes are arranged on the x-axis by expression level, and the mean gene count (counts per million) value is shown on the y-axis. Gene symbols for all of the genes could not be shown on the x-axis. Data from patients from the low As exposure group are shown on the left (**A**), and data from patients from the high As exposure group are shown on the right (**B**). The average magnitude of overexpression was significantly higher (*p* = 1.30 × 10^−12^) if the patient had low As exposure [5.41-fold change (95% CI 4.06–7.20)] compared to those with high As exposure [1.53-fold change (95% CI 1.25–1.86)).

**Figure 6 cancers-14-05598-f006:**
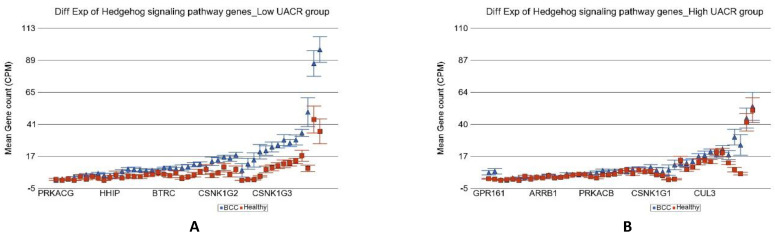
Differential gene expression of Hedgehog signaling pathway genes in BCC tissue (in blue) compared to healthy skin tissue (in red). Genes are arranged on the x-axis by expression level, and the mean gene count (counts per million) value is shown on the y-axis. Gene symbols for all of the genes could not be shown on the x-axis. Data from patients from the low As exposure group are shown on the left (**A**), and data from patients from the high As exposure group are shown on the right (**B**). The average magnitude of overexpression was significantly higher (*p* = 1.40 × 10^−7^) if the patient had low As exposure [4.34-fold change (95% CI 3.30–5.71)] compared to those with high As exposure [1.78-fold change (95% CI 1.47–2.14)].

**Figure 7 cancers-14-05598-f007:**
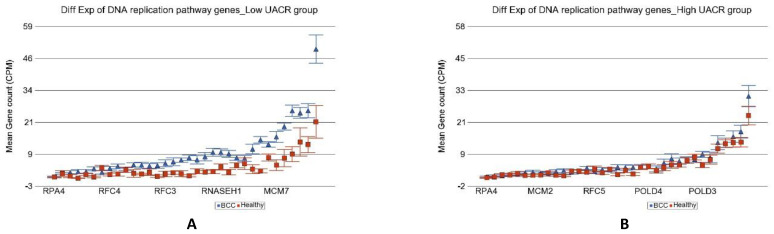
Differential gene expression of DNA replication pathway genes in BCC tissue (in blue) compared to healthy skin tissue (in red). Genes are arranged on the x-axis by expression level, and the mean gene count (counts per million) value is shown on the y-axis. Gene symbols for all of the genes could not be shown on the x-axis. Data from patients from the low As exposure group are shown on the left (**A**), and data from patients from the high As exposure group are shown on the right (**B**). The average magnitude of overexpression was significantly higher (*p* = 4.29 × 10^−9^) if the patient had low As exposure [4.42-fold change (95% CI 3.07–6.35)] compared to those with high As exposure [1.17-fold change (95% CI −1.09–1.51)].

**Figure 8 cancers-14-05598-f008:**
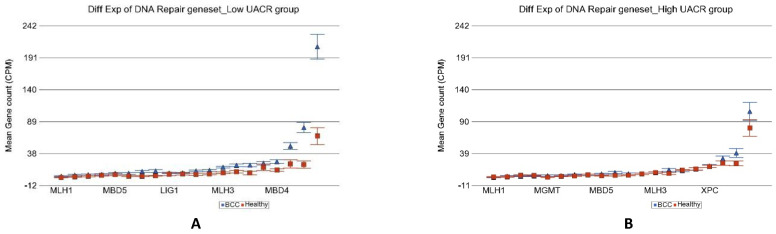
Differential gene expression of DNA repair genes in BCC tissue (in blue) compared to healthy skin tissue (in red). Genes are arranged on the x-axis by expression level, and the mean gene count (counts per million) value is shown on the y-axis. Gene symbols for all of the genes could not be shown on the x-axis. Data from patients from the low As exposure group are shown on the left (**A**), and data from patients from the high As exposure group are shown on the right (**B**). The average magnitude of overexpression was significantly higher (*p* = 1.84 × 10^−6^) if the patient had low As exposure [2.79-fold change (95% CI 1.98–3.93)] compared to those with high As exposure [1.01-fold change (95% CI −1.26–1.27)].

**Figure 9 cancers-14-05598-f009:**
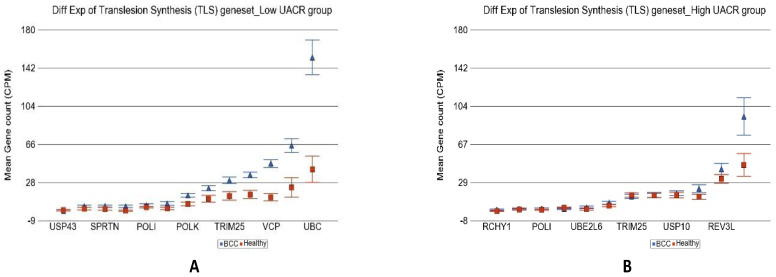
Differential gene expression of translesion synthesis (TLS) genes in BCC tissue (in blue) compared with healthy skin tissue (in red). Genes are shown on the x-axis, and the mean gene count (counts per million) value is shown on the y-axis. Data from patients from the low-As-exposure group are shown on the left (**A**), and data from patients from the high-As-exposure group are shown on the right (**B**). The average magnitude of overexpression was significantly higher (*p* = 2.68 × 10^−5^) if the patient had low As exposure (3.67-fold change (95% CI 2.34–5.74) compared with those with high As exposure (1.14-fold change (95% CI −1.19–1.55)).

**Figure 10 cancers-14-05598-f010:**
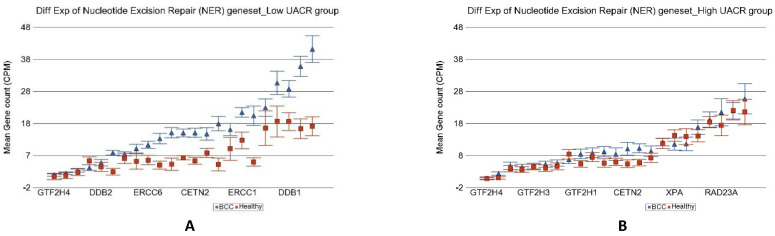
Differential gene expression of nucleotide excision repair (NER) genes in BCC tissue (in blue) compared to healthy skin tissue (in red). Genes are shown on the x-axis, and the mean gene count (counts per million) value is shown on the y-axis. Data from patients from the low-As-exposure group are shown on the left (**A**), and data from patients from the high-As-exposure group are shown on the right (**B**). The average magnitude of overexpression was significantly higher (*p* = 1.61 × 10^−4^) if the patient had low As exposure [2.35-fold change (95% CI 1.71–3.24)] compared to those with high As exposure [1.12-fold change (95% CI −1.12–1.39)].

**Figure 11 cancers-14-05598-f011:**
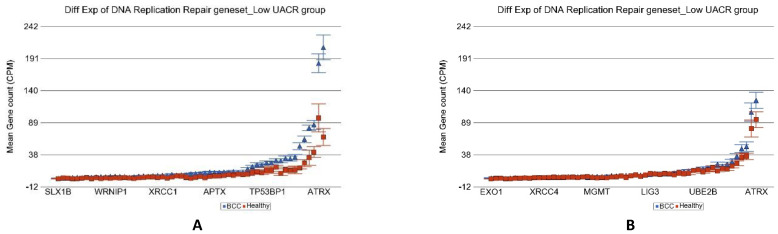
Differential gene expression of replication stress site (RSS) genes in BCC tissue (in blue) compared to healthy skin tissue (in red). Genes are shown on the x-axis, and the mean gene count (counts per million) value is shown on the y-axis. Data from patients from the low-As-exposure group are shown on the left (**A**), and data from patients from the high-As-exposure group are shown on the right (**B**). The average magnitude of overexpression was significantly higher (*p* = 2.28 × 10^−11^) if the patient had low As exposure [3.55-fold change (95% CI 2.78–4.53)] compared to those with high As exposure [1.29-fold change (95% CI 1.09–1.52)].

**Figure 12 cancers-14-05598-f012:**
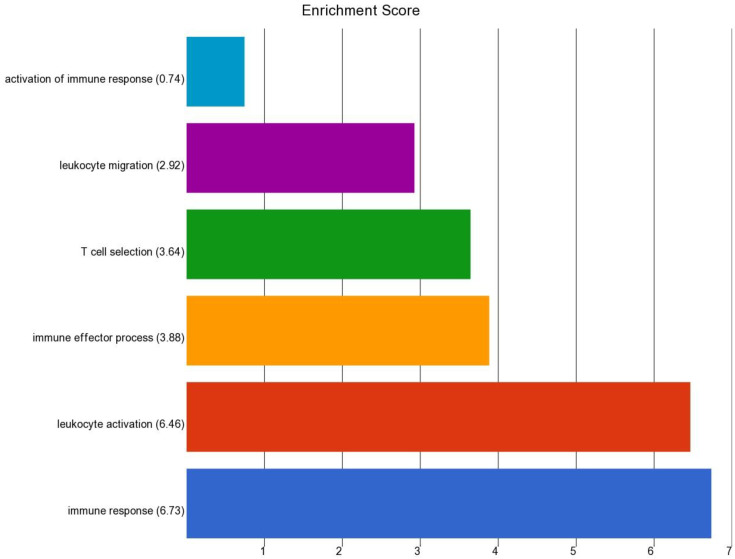
Enrichment score of the list of genes that were significantly downregulated in BCC with high As exposure compared to BCC with lower As exposure.

**Figure 13 cancers-14-05598-f013:**
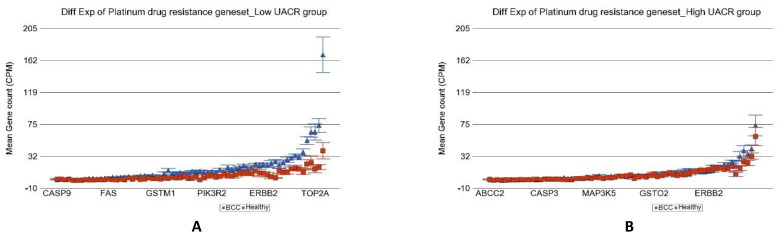
Differential gene expression of platinum drug resistance genes in BCC tissue (in blue) compared to healthy skin tissue (in red). Genes are shown on the x-axis, and the mean gene count (counts per million) value is shown on the y-axis. Data from patients from the low-As-exposure group are shown on the left (**A**), and data from patients from the high-As-exposure group are shown on the right (**B**). The average magnitude of overexpression was significantly higher (*p* = 4.10 × 10^−19^) if the patient had low As exposure [4.18-fold change (95% CI 3.34–5.23)] compared to those with high As exposure [1.20-fold change (95% CI 1.02–1.40)].

**Table 1 cancers-14-05598-t001:** Differential Expression of Cancer-Related Gene Sets in BCC Tissue Compared to Healthy Tissue in Low Arsenic Exposure Group and High Arsenic Exposure Group.

Stratification	Interaction *p*	Low UACR			High UACR		
		Fold Change	(95% CI)	*p*	Fold Change	(95% CI)	*p*
**DNA Repair**	1.84 × 10^−6^	2.79	(1.98–3.93)	7.08 × 10^−9^	1.01	(−1.26–1.27)	0.96
Growth Factor Receptors	6.16 × 10^−3^	3.44	(1.51–7.84)	3.64 × 10^−3^	−1.19	(1.49–−2.09)	0.55
Pro-Apoptosis	2.79 × 10^−2^	8.37	(2.54–27.60)	5.74 × 10^−4^	1.65	(−1.38–3.75)	0.23
Caspases Initiator	3.22 × 10^−2^	4.44	(1.68–11.75)	2.98 × 10^−3^	1.22	(−1.60–2.39)	0.55
Anti-TSG	4.43 × 10^−2^	10.21	(3.75–27.77)	9.11 × 10^−6^	2.94	(1.48–5.86)	2.29 × 10^−3^
Anti-Apoptosis	0.07	3.84	(1.33–11.10)	1.35 × 10^−2^	1.19	(−1.75–2.46)	0.64
Tumor Suppressor Gene	0.13	3.42	(1.65–7.13)	1.13 × 10^−3^	1.72	(1.04–2.85)	3.51 × 10^−2^
Caspases Executioner	0.16	4.10	(1.21–13.88)	2.35 × 10^−2^	1.42	(−1.63–3.29)	0.41

**Table 2 cancers-14-05598-t002:** Differential Expression of DNA-Damage-Related Gene Sets in BCC Tissue Compared to Healthy Tissue in Low Arsenic Exposure Group and High Arsenic Exposure Group.

Stratification	Interaction *p*	Low UACR			High UACR		
		Fold Change	(95% CI)	*p*	Fold Change	(95% CI)	*p*
**Translesion Synthesis (TLS)**	2.68 × 10^−5^	3.67	(2.34–5.74)	2.00 × 10^−8^	1.14	(−1.19–1.55)	0.41
Fanconi Anemia (FA)	4.22 × 10^−5^	5.82	(3.09–10.97)	7.22 × 10^−8^	1.16	(−1.33–1.80)	0.50
Nucleotide Excision Repair (NER)	1.61 × 10^−4^	2.35	(1.71–3.24)	1.81 × 10^−7^	1.12	(−1.12–1.39)	0.33
Base Excision Repair (BER)	5.67 × 10^−4^	3.19	(1.91–5.32)	1.07 × 10^−5^	1.07	(−1.33–1.52)	0.72
Direct Reversal Repair (DRR)	1.01 × 10^−2^	11.41	(2.65–49.16)	1.28 × 10^−3^	1.10	(−2.47–3.01)	0.85
Microhomology mediated end joining (MMEJ)	1.82 × 10^−2^	3.48	(1.43–8.46)	6.16 × 10^−3^	−1.05	(1.75– −1.94)	0.87
Homologous Recombination (HR)	3.00 × 10^−2^	2.57	(1.64–4.03)	3.97 × 10^−5^	1.41	(1.04–1.92)	2.92 × 10^−2^
Non-homologous end joining (NHEJ)	3.39 × 10^−2^	3.34	(1.81–6.14)	1.23 × 10^−4^	1.50	(−1.01–2.28)	0.06
Mismatch Repair (MMR)	3.97 × 10^−2^	2.67	(1.47–4.85)	1.34 × 10^−3^	1.25	(−1.21–1.88)	0.29
Checkpoint Signaling	0.051	2.93	(1.58–5.43)	7.12 × 10^−4^	1.39	(−1.10–2.12)	0.13

**Table 3 cancers-14-05598-t003:** Differential Expression of RSS-Related Gene Sets in BCC Tissue Compared to Healthy Tissue in Low Arsenic Exposure Group and High Arsenic Exposure Group.

Stratification	Interaction *p*	Low UACR			High UACR		
		Fold Change	(95% CI)	*p*	Fold Change	(95% CI)	*p*
**RNA Processing**	9.27 × 10^−20^	2.56	(2.19–2.98)	1.69 × 10^−31^	1.06	(−1.05–1.18)	0.27
Protein Translation	1.26 × 10^−14^	3.05	(2.46–3.78)	1.54 × 10^−23^	1.09	(−1.07–1.26)	0.28
Chromatin TF Transcription	1.03 × 10^−13^	3.51	(2.78–4.41)	7.29 × 10^−26^	1.21	(1.03–1.41)	2.03 × 10^−2^
DNA Replication Repair	2.28 × 10^−11^	3.55	(2.78–4.53)	9.75 × 10^−24^	1.29	(1.09–1.52)	3.22 × 10^−3^
Cell Survival	2.79 × 10^−8^	2.81	(2.16–3.66)	2.90 × 10^−14^	1.13	(−1.06–1.36)	0.19
Immune Regulation	1.65 × 10^−5^	3.37	(2.36–4.80)	2.96 × 10^−11^	1.31	(1.02–1.67)	3.18 × 10^−2^
Cell Cycle	4.47 × 10^−5^	3.14	(2.35–4.20)	2.49 × 10^−14^	1.50	(1.23–1.84)	6.72 × 10^−5^
GF Signaling	1.26 × 10^−4^	2.76	(1.84–4.15)	1.13 × 10^−6^	1.05	(−1.26–1.39)	0.72
Cell Movement	3.54 × 10^−4^	2.29	(1.74–3.00)	3.35 × 10^−9^	1.25	(1.04–1.51)	1.87 × 10^−2^
Stress Responses	2.15 × 10^−3^	4.44	(2.60–7.59)	7.44 × 10^−8^	1.61	(1.11–2.32)	1.16 × 10^−2^
Metabolism	1.95 × 10^−2^	3.70	(1.79–7.64)	4.52 × 10^−4^	1.30	(−1.27–2.14)	0.30
Development Regulation	2.04 × 10^−2^	2.26	(1.31–3.89)	3.37 × 10^−3^	1.04	(−1.40–1.51)	0.85
Angiogenesis	0.06	4.87	(1.53–15.48)	7.72 × 10^−3^	1.30	(−1.71–2.87)	0.52

## Data Availability

All the supporting data are presented in the tables presented in the main manuscript and as additional Appendix A.

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
