# Peer review of "Interaction of Arsenic Exposure and Transcriptomic Profile in Basal Cell Carcinoma"

_cancers, 2022, doi:10.3390/cancers14225598_

Round 1

Reviewer 1 Report

The manuscript by Kibriya et al. examined gene expression profiling in basal cell carcinoma (BCC) by transcriptome wide RNA sequencing after As exposure. They found that many of the gene pathways were involved in BCC and that high arsenic exposure was associated with impaired (a) DNA replication, (b) cellular response to DNA damage repair and (c) immune response.  

The major conclusions of this research are justified by the results. The results of this work may be worth publishing. However, the study requires improvement in some aspects.

Major revision:

1. The results of basal cell carcinoma should be showed by pathological biopsy before RNA-Seq.

2. Some key differentially significant genes should be further verified by qPCR after the sequencing data analysis in BBC and BBC after As exposure.

3. The content of As should be determined by high performance liquid chromatography in the low and high As exposure.

Reviewer 2 Report

The topic of this Paper is very interesting, however it has many limitations.

1. The technics are poorly described; there is no information’s about which type of biopsy material was used.

Is it correct that a total of 26 tumors vs. 16 other patients were used. The Paper would be by far more important if the BCC’s are combined to normal skin of the same individual. 

2. The Bioinformatics Analysts must be checked by an expert.

3. The Authors should do a similar analysis on BCC material in a control population. Over all i think that numbers of samples is rather small for conclusive and statistical significance results.

Round 2

Reviewer 1 Report

no problem

Reviewer 2 Report

none